An interactive and integrated framework for collaborative product development in cloud manufacturing using STEP standard-based ontology model

Zaringhalam SeyedehTina 1
Khalilzadeh Mohammad 2
http://orcid.org/0000-0001-7087-6946 Fatahi Valilai Omid 3 ofatahivalilai@constructor.university
1 Department of Industrial Engineering, Science and Research Branch, Islamic Azad University , Tehran, Tehran , Iran
2 CENTRUM Católica Graduate Business School, Pontificia Universidad Católica del Perú , Lima , Peru
3 School of Business, Social & Decision Sciences, Constructor University , Bremen, Bremen , Germany
Cambronero M Emilia
Electronic publication date: 2023 Aug 29
Publication date: 2023
Volume: 9
Electronic Location ID: e1530
Received 2022 Oct 31; Accepted 2023 Jul 20
Copyright: © 2023 Zaringhalam et al.
Copyright year: 2023
Copyright holder: Zaringhalam et al.
License: This is an open access article distributed under the terms of the Creative Commons Attribution License, which permits unrestricted use, distribution, reproduction and adaptation in any medium and for any purpose provided that it is properly attributed. For attribution, the original author(s), title, publication source (PeerJ Computer Science) and either DOI or URL of the article must be cited.
License URL: https://creativecommons.org/licenses/by/4.0/

Keywords: Cloud manufacturing, Integration, STEP standard, Interactive product development

Funding: The authors received no funding for this work.

==============================
Due to the growing competition of firms in manufacturing industries, the enhancement of the design of new products for quick and collaborative interaction of stakeholders is inevitable. A customer-oriented approach during the enhancing of product design requires the support of various stakeholders’ collaboration from their functionality perspectives. Accordingly, manufacturing firms try to increase the product quality through the continuous collaboration and speeding up of the design process. Moreover, new manufacturing paradigms like cloud manufacturing require global and distributed collaboration of stakeholders. One competitive key necessity for firms is the integration of stakeholders in the design and production stages. This article has proposed an interactive and integrated framework for a cloud collaborative manufacturing system. The proposed framework uses the international STEP standard which ensures data integrity in the whole product development processes. The framework enables stakeholders to work collaboratively and interactively in their disciplines, while it also ensures coherency of modifications to avoid conflicts. The proposed model is introduced using an improvisational design method based on the STEP standard to create an interactive platform with a service-oriented approach and supports stakeholders for their own semantics. In addition to examining the needs of an interactive system, the article provides an architecture for implementation of this perspective. A comprehensive case study is designed to show the capabilities of the proposed framework.

Introduction

Nowadays, firms are exposed to changes by information technology (IT) and the related merits of the digital transformation which affect all areas of the supply chain operations from the contact with suppliers to product development and production processes and interacting with customers (Mohammadi, Mukhtar & Peikari, 2011). Firms’ product engineering unit interaction with suppliers is one of those fields which is believed to be affected highly by this transformation (Kumar & Rodrigues, 2020). This unit is the key for promoting innovation to real products within the firms’ operations and the supply chain (Kumar & Rodrigues, 2020). One of the dominant proposed solutions is to transform the traditional supply chains into a cloud-based supply chain (Morgan & Conboy, 2013). In addition, the increasing demand for new products coupled with market globalization are key issues for firms involved collaboratively in the product development processes (Rajapathirana & Hui, 2018). Therefore, a collaborative production design and development environment is inevitable for nowadays production firms.

Traditional product data exchange has mainly focused on sharing geometric data, which includes data elements like the shape, size, and physical properties of a product. However, with the emergence of model-based definition (MBD), there has been a shift towards sharing non-geometric information as well. MBD is an approach to product design and manufacturing that uses a 3D model as the authoritative source of information for all aspects of a product. Non-geometric information that is now being shared as part of MBD includes product manufacturing information (PMI) and design intent (Jian et al., 2018). So, one of the main issues in collaborative product development is the seamless data communication among agents engaged in parallel disciplines (Hu, Zhou & Li, 2010; Zhang et al., 2014). Although there have been efforts in several cloud computing related solutions to facilitate data sharing and collaboration in product development disciplines like knowledge mapping, merging, searching, and transferring in product design procedure (Bohlouli, Holland & Fathi, 2011), within international standards for transmission of CAD data there are still challenges. Moreover, data communication problems often arise when two or more CAD/CAM/CAE systems work together to form a design or process plan. To solve this problem, it is necessary to translate the product data of one system in a way that another system can use it without violating the other stakeholder modifications (Marjudi et al., 2010). This needs a detecting mechanism to use both a standard data model for the transmission of CAD design data and also an ontology marked model which encompass the theme of cooperation through the semantics between designers (Weber & Anderl, 2021; Tessier & Wang, 2013; Perzylo et al., 2015).

This article tries to solve this problem by providing an interactive and integrated platform for collaborative design based on cloud service. Therefore, the proposed platform uses the STEP standard. STEP is based on the Product Data Exchange Specification (PDES), which is an attempt to define data designed to improve interoperability between manufacturing companies as a standard for exchanging data formats used to communicate a plan (Valilai & Houshmand, 2010). Also, Step-Based Feature is considered as an essential tool for integrating CAD systems. In this way, data integrity is maintained. It also provides an ontology-based solution to control interactions at the level of collaborative design of parts. By using this method, instead of requiring companies to use a specific method; they will have a service-oriented protocol for information mapping that prevents interference. In addition, for the usability of the proposed architecture in different geographical locations, the platform has been implemented in a service-oriented and cloud-based manner. Also, the storage of all work steps and history of activities can be retrieved and provides data availability at any time and place. This article presents a STEP-based feature modeler (STEP-FM) for prismatic components. High-level 3D solid features are used as basic entities to design parts. The modeler selects the base shape of the part and the overall size and provides the information needed to define the feature size, position, orientation, and other features. The designed part is then exported as a STEP XML data format. The proposed method and architecture for product information exchange are based on XOEM (XML-STEP) and OWL, which enables the resolution of issues related to heterogeneous workspaces, systems, and information. This approach is designed to support dynamic and agile requirements (Jian & Ying, 2007).

The article organizes as follows: In ‘Literature Review’, the proposed literature and related projects are reviewed. A proposed integrated and common platform is introduced in ‘Integrated Platform to Support Collaborative Distributed Production System Based on a Service-Oriented Approach and its Cloud Computing Model’. In ‘Illustrative Cases’, the proposed design is redefined based on the service-orientation approach and cloud computing model, and a brief case study is described in ‘Discussion’.

Literature review

The basic requirements of an integrated collaborative distributed production environment are described here.

Distributed support structures and procedures

Smart Manufacturing

Smart manufacturing is a broad category of manufacturing that employs Computer-Integrated manufacturing, high levels of adaptability in design changes, digital information technology, and more flexible technical workforce training; Other goals sometimes include rapid changes in demand-based production levels (Davis et al., 2012), supply chain optimization, efficient production, and recycling (Shipp et al., 2012). The broad definition of smart manufacturing encompasses many different technologies. Some of the key technologies include big data processing capabilities, advanced robotics, cloud generation, and artificial intelligence production.

Using the capabilities of the Internet, manufacturers are able to increase the integration and storage of data. Using cloud technology allows companies to access many computing resources. This allows servers, networks, and other storage applications to be created and distributed quickly (Yao et al., 2017; Lu & Xu 2017).

Cloud computing

As computers became more common, scientists provided users with ways to build computing power on a large scale by sharing resources (Velte, Velte & Elsenpeter, 2019). However, the availability of high-capacity networks, low-cost computers, and storage devices, as well as the widespread use of hardware virtualization, service-oriented architecture, and standalone computing, have led to the growth of cloud computing (Knorr & Gruman, 2008).

Cloud manufacturing (CMfg) is a new production model that uses advanced virtualization and service-oriented production models. CMfg is a kind of parallel, networked, and distributed system that consists of an integrated and interconnected virtual service set with production and intelligent management capabilities that use service demand to provide solutions throughout the production life cycle (Schaefer et al., 2012).

STEP-based integration

In design and manufacturing, many systems are used to manage product technical data. Each system has its own data formats, so the same information must be entered multiple times in multiple systems, leading to redundancy and errors. This problem is not unique to manufacturing but is more acute in the design phase because the design data is complex and three-dimensional, leading to an increased scope for errors and misunderstandings between operators (Houshmand & Valilai, 2013). STEP is based on Product Data Exchange Specification (PDES), which began in the mid-1980s and was submitted to ISO in 1988 (Kutz, 2002). However, PDES is an attempt to define data designed to improve interoperability between manufacturing firms to improve productivity (Powers, 2003). Interaction in manufacturing sectors in different geographical locations in the production process leads to quality improvement and cost reduction. A major obstacle is the lack of a common language. ISO10303 standard solution is one of the best ways to integrate CAx information. STEP (Standard for Product Data Interchange) written using the EXPRESS modeling language is a CAD file format that can be used among various CAD programs such as CATIA, Creo, SolidWorks, NX, Inventor, etc. Jian & Meng (2011) presented the XOEM+OWL semantic model, which is based on ontology and the STEP standard. The XOEM component is an XML-based data model for representing STEP, while the OWL component provides a semantic layer in the form of a pattern graph. This model enables the integration of product design, but it does not provide a solution for concurrent collaborative design in terms of flagging or marking the changes among collaborative product development disciplines and enabling a conflict resolution framework (Jian & Meng, 2011).

Hu, Zhou & Li (2010), Zhang et al. (2014), a network-based network production model introduced with a new service called cloud generation aimed at solving more complex manufacturing problems and conducting large-scale collaborative production which is inspired by cloud computing. Liu et al. (2019) showed that the goals of cloud computing are mainly to facilitate sharing and collaboration.

STEPs are used as data formats to transfer designs to a different system. Platforms developed by STEP standards use the standard to exchange a design (Ho, Tsuei & Yang, 1988). But one of the problems in this direction, is the simultaneous changes of the designers on the piece. This standard is detailed in several Application Protocols (APs). Among them, we can refer to AP 203, AP 214 and AP 242 protocols related to CAD mechanical design, 3D dimensional geometry and product information. The AP 203 protocol defines the topology and configuration data management of solid models for mechanical parts and assemblies. The AP 214 protocol includes all the features of AP 203 plus colors, layers, GD&T (geometric dimensions and tolerances). The AP 242 protocol includes all the features of AP 203 and AP 214. In addition to those related issues PMI also supports 3D semantics, 3D parametric/geometric constraint design, access and digital rights management (DRM), and long-term archiving. Therefore, considering that AP242 combines both AP 203 and AP 214. It is considered as the preferred protocol in the proposed architecture of this article. The use of the semantic visualization model of STEP ontology knowledge base enables the integration of STEP entity object descriptions and the definition of semantic links and mappings between them (Jian, Pang & Tao, 2014). The article examines the need for a participatory distributed design and production.

An overview on integrated and collaborative distributed production system considering product data integration

The article has demonstrated the research gap in Table 1. The main literature review gaps are based on capabilities of research studies for supporting:

Table 1 Analysis of focus domains of the literature research studies.

Reference	Cloud
manufacturing
support	Collaborative manufacturing
environment	Integration based on STEP standard	Real time information exchange and sharing	
Tao et al. (2011)	YES	YES			
Jian & Meng (2011)	YES		YES	YES	
Helu & Hedberg (2015)		YES			
Shen et al. (2007)		YES			
Osório, Afsarmanesh & Camarinha-Matos (2011)	YES	YES		YES	
Xia et al. (2016)	YES	YES			
Valilai & Houshmand (2013)	YES	YES	YES		
Siller et al. (2009)	YES	YES	YES		
Abedi et al. (2016)	YES	YES			
Toquica, Alvares & Bonnard (2018)	YES		YES		
Ramnath et al. (2020)		YES	YES		
Ding et al. (2022)		YES		YES	
Göppert et al. (2021)		YES		YES	
This article	YES	YES	YES	YES	

Service oriented characteristics of cloud manufacturing environment.

Enabling the collaboration of product developers in design phase and avoiding the conflicts of product design modification by a concrete product design ontology model

Fulfilling the requirements of data integration based on STEP standard while enabling the collaboration of product designers

Supporting the access to product ontology model for modification to collaborators

Table 1 shows the gap for an effective solution to fulfill the above-mentioned requirements. Tao et al. (2011) propose a platform for collaborative production that is based on cloud manufacturing, however, it does not use the STEP standard for data integration and also the model lacks a procedure for cooperation between the manufacturing of agents in distributed environments. Moreover, they do not consider the simultaneous design control problems. Helu & Hedberg (2015) proposes a model that addresses only the integration aspect of manufacturing and does not address issues such as the STEP standard for collaborative and the use of cloud technology. The proposed platform by Shen et al. (2007) addressed the issue of integrated co-production but lacks a solution for co-design and co-production and a standardization method for co-production as well. The design proposed by Osório, Afsarmanesh & Camarinha-Matos (2011) introduces co-production in the cloud computing platform. In addition, in their paper, concurrence in participatory production has been discussed. The proposed platform is also integrated and distributed but does not support the STEP-based platform.

Xia et al. (2016) proposes a platform to enable product data integration and co-production, however, the approach does neither ensure product data integration according to the STEP standard, nor does it address the issue of co-production in co-production, although cloud manufacturing has not been used as a platform. In their platform, Valilai & Houshmand (2013) and Siller et al. (2009) state the possibility of co-production in integrated production and provide a basis for distributed manufacturing using a cloud computing method. The use of the STEP standard ensures integration and collaborative production at an acceptable level. However, the problem that have left unanswered is the simultaneous discussion of design and production partnerships. The model proposed by Siller et al. (2009), using the STEP standard for integration and participation in manufacturing, also addresses the issue of decentralization and distributed production. However, in this research, the cloud manufacturing platform has not been used and the issue of concurrence in design and development in the discussion of co-production has remained unanswered.

The purpose of Abedi et al. (2016) enhance collaboration and improve integration by using cloud computing. However, in this article, STEP standard and Real-Time Information Sharing have not been used. In the research of Toquica, Alvares & Bonnard (2018), a solution is proposed based on the STEP-NC standard combined with a suitable CAD/CAM solution. A platform has been developed to enable advanced and intelligent manufacturing with the possibility to integrate several modules of simulation, cloud manufacturing. However, the problem that has been left unanswered is the simultaneous discussion of real-time design and co-production. Ramnath et al. (2020) proposes a platform that addresses the issue of integrated co-production by using the STEP standard. but it does not use cloud technology and lacks a solution for Real-Time designing.

An integrated platform to support collaborative distributed production system based on a service-oriented approach and its cloud computing model

The proposed multilayer approach based on the integration of STEP-based product data is shown in Fig. 1. Adopting the proposed platform can overcome the existing shortcomings and limitations, as well as allow real-time design and development among developers. In addition, the proposed platform supports different products based on their own product data structure. Finally, structural layers are designed to overcome inconsistencies and facilitate the integration and collaboration of globally distributed production.

Figure 1 Proposed architecture.

The article presents the architecture as a service-oriented approach based on the STEP standard.

The proposed service-oriented approach uses an Extensible Markup Language (XML), which was widely used by many distributed computing projects after the creation of XML by the W3C in 1996; XML is then converted to RDF and finally to OWL, the conversions are based on the language bank. The relationship involves the transfer of product data, and product data is interpreted semantically according to the STEP standard. Designers can design the desired product development using the real-time management mechanism considered in the architecture. After finalizing the design for implementation through IoT technology, the design is provided to the machines for execution.

In the following, the proposed architecture layers are examined:

1) Interface Layer

The layer contains forms for users to connect to the system and can send CAx software packages through the interface to select the desired data format. The layer allows sending and receiving product data.

2) Mapping Layer

In this layer, the data received in the “interface layer” is translated into XML, RDF and Owl languages using the translation rules in the “Rules section”, respectively. To convert data from CAx formats to XML and vice versa. A cloud service is used for XMLization/DeXMLization. However, the service uses XML_STEP Mapping to convert the format. The mapping consists of two parts: the first part includes inferring product data based on XML structure and the second part is based on STEP modules. Thus, product data in various formats is converted to STEP-based data structure in XML format. Then, STEP XML data is converted to Resource Description Framework (RDF). At this level, a data model is created for storage and retrieval which can be processed by the machine. The data is then converted to Web Ontology Language (OWL) to model relevant knowledge and prepare for the semantic interpretation level.

3) Logical Interpreter Layer

Different product data is delivered to the layer after passing the mapping stage, after entering this level, the data flag is checked in and goes to the semantic interpreter section. At this level, product data is exchanged seamlessly. While exchanging data, other applied changes will be reviewed and in case of discrepancies, in the changes, the designers of the interfering sections will be notified. After removing the flag interference, the data is checked out and the product data is delivered to the Store/Restore layer for storage or retrieval.

4) Store/Restore Layer

The layer is designed to store and restore information from the database, designing different parts of the database is based on standard STEP modules. This provides an integrated structure for storing product data. After finalizing the design, this layer sends information by IoT technology to the relevant manufacturing machines for production based on the final design and design modules.

Illustrative cases

Case study

A case study has been considered to evaluate the feasibility of the proposed architecture. In the case study, a hypothetical decentralized company is considered, in which design and manufacturing agents interact interactively on components in remote environments. Each of these units uses a CAx software package for design. The exchange of product information for interaction is based on the proposed architecture: The solution provides an integrated hospital based on the STEP standard. The platform, however, is provided as a cloud service to design units in different geographical locations.

Design

The snippet is designed using a CAD software package. The designed snippet is shown in Fig. 2 below and the detail of all the case studies is accessible from http://doi.org/10.6084/m9.figshare.21401262. The text snippet design is based on STEP standard rules. These steps are conducted in the architecture interface section.

Figure 2 The designed piece isometric view.

Mapping

After the first step, the design file enters the mapping section.

For this purpose, the software is designed in three parts as shown in Fig. 3 (the developed codes can be accessible from http://doi.org/10.6084/m9.figshare.21401163):

Figure 3 Designed interface for serialization of data.

– Convert STP file to XML

– Convert XML file to OWL

– Convert STP file to OWL

This software, which is designed for the web, can be accessed in the cloud. In this way, each developer will be able to convert their file by uploading their changes file in this tool. Finally, the project ontology is created using the owl file, and according to the defined rules, the developer design interactions are identified.

In the first step of mapping, using XML rules, the piece information is placed in xml format as illustrated in Table 2. An example of the xml code of the snippet can be seen below.

Table 2 The sample of XML code snippet.

XML Code	
<entity_id>#5<entity_name>”Product”</entity_name><attribute>	
<id type=”string”>Part1</id>	
<name type=”string”>’’</name>	
<description type=”string”>’’</description>	
<frame_of_reference type=”ENTITY”>	
<SET><entity_instance_refrefid>#2</entity_instance_refrefid></SET>	
</frame_of_reference></attribute></entity_id>	
<entity_id>#47<entity_name>”CARTESIAN_POINT”</entity_name><attribute>	
<name type=”string”>’’</name>	
<coordinates type=”real”>	
<real_literal>0</real_literal>	
<real_literal>0</real_literal>	
<real_literal>0</real_literal>	
</coordinates></attribute></entity_id>	
<entity_id>#94<entity_name>”DIRECTION”</entity_name><attribute>	
<name type=”string”>”Axis2P3D Direction”</name>	
<direction_ratios type=”real”>	
<real_literal>0</real_literal>	
<real_literal>1</real_literal>	
<real_literal>0</real_literal>	
</direction_ratios>	
</attribute></entity_id>	
…	
</express_data>	
</iso_10303_21>	

STEP to XML conversion is conducted automatically with the built-in converter. The STEP-based feature modeler is a powerful tool for creating complex 3D models using a feature-based approach. The above 3D solid features are used as basic entities for part design. Information, including the basic shape and overall size of the part, provides the information needed to determine the size, position, orientation, and other characteristics of the feature, which is created in the STEP XML (ISO-based) format. Also, the object-oriented method has been used in the definition and implementation of the product model. Table 3 illustrates a part of the STEP to XML converter.

Table 3 Sample of STEP to XML converter code.

Convertor STEP To XML	
protected void Convert_Click(object sender, EventArgs e) {	
try {	
string DataFirst = “DATA;”, DataLast = “ENDSEC;”;	
string MainData, WorkData, Strand;	
string Result,FinalResult = “”;	
int startIndex, EndIndex, SubIndex;	
if (xmlString.Text != “”) {	
string InputData = xmlString.Text;	
startIndex = InputData.IndexOf(DataFirst) + 7;	
EndIndex = InputData.IndexOf(DataLast);	
MainData = InputData.Substring(startIndex, EndIndex – startIndex);	
/////////////////	
btnConvert.Enabled = false;	
WorkData = MainData;	
	
while ((SubIndex = WorkData.IndexOf(“\r\n”)) != −1) {	
Strand = WorkData.Substring(0, SubIndex);	
Result = MainMaker(Strand);	
WorkData = WorkData.Substring(SubIndex + 1, MainData.Length – SubIndex – 1);//	
FinalResult += Result;}	
TextBox2.Text = FinalResult;} }	
catch (Exception ex) {theDiv.Visible = true;}}	
//////////////////////////////////////////////////////////	
string MainMaker(string Strand) {	
string[] Result = new string[100];	
string[] SetResults = new string[100];	
string FinalResult;	
string s = Strand;	
string FinalOutput = “”;	
string Output = “”;	
int h , j = 0;	
//////	
int startEntity = s.IndexOf(“=”) + 1;	
int EndEntity = s.IndexOf(“(“, startEntity);	
	
string ResultEntity = s.Substring(startEntity, EndEntity – startEntity);	
//////	
int StartID = s.IndexOf(“#”) + 1;	
int EndID = s.IndexOf(“=”, StartID);	
string ResultID = s.Substring(StartID, EndID – StartID);	
/////	
int start = s.IndexOf(“(“) + 1;
int end = s.IndexOf(“) ;”, start) – 1;	
string Edate = s.Substring(start, end – start + 1);	
string[] arrayStr = Edate.Split(‘,’);	
int len = arrayStr.Length;	
for (int development = 0; i <= len–- 1; i++) {	
string PEData = ParseData(eDate, i);	
h = i;	
if (PEData.Contains“"(”"))	
{int Istart = eDate.IndexOf“"“") + 1;	
int Iend = eDate.IndexOf“"”", Istart)–- 1;	
string iData = eDate.Substring(Istart, Iend–- Istart + 1);	
string[] arrayStrb = iData.Split‘'’');	
int Ilen = arrayStrb.Length;	
for (j = 0; j <= Ilen–- 1; j++)	
{string PIData = ParseData(iData, j);	
SetResults[j] = PIData;Output +=“"<entity_instance_refrefid >“"+ SetResults[j] +“"</entity_instance_refrefid”";	
h++;	
}	
}	
if (i == h) {Result[i] = PEData.Replace“"“",“”").Replace“"”",“”");} else {Result[i] = Output;}	
///////////////////////////////////////CARTESIAN_POINT///////////////////////////////////////////	
if (ResultEntity ==“"CARTESIAN_POIN”") {	
FinalOutput =“"< attribute > < name type = string >“" +Result[0] +“"</name >< coordinates type = real >	
< real_literal ”" +Result[1]+“"</real_literal >< real_literal ”" +Result[2]+“"</real_literal >< real_literal >“"	
+Result[3]+“"</real_literal ></coordinates ></attribute”";}	
/////////////////////////////////////////////////////	
FinalResult =“"< entity_id >”" + ResultID +“"< entity_name ”" + ResultEntity +“"</	
entity_name ”" + FinalOutput +“"</entity_id ”";	
return FinalResult;	

Store

As the code is converted to XML, the information is stored in a relational database. Figure 4 demonstrates the designed to store database information based on the entity diagram. In this way, it is possible to identify the relationships to the entities and to show the parent-child relationship. This provides a level of control when designing a partnership to identify interferences. The relational database also stores information and also has the ability to retrieve all previously stored information for each entity. This allows the information to be returned in the event of any errors or possible interference.

Figure 4 Snippet database view.

In addition, by creating a table, equivalent to history, both developers and designers can access the previous information of each of the required entities based on the date and time or the version they have defined. In addition, it is possible to compare changes with the previous version at the same time. The stored changes are based on the ID of each designer, and this privilege allows a designer to review the actions of other designers or send a message to each of the developers.

After creating STEP XML, the file is ready for ontology mapping as shown in Fig. 5. For ease of mapping two converters, designed from STP to OWL format and from XML to OWL format. All entities are defined as classes and their relationships in the ontology.

Figure 5 The snippet information ontology.

A sample of STP to OWL converter code has been also developed besides STP to OWL converter code converter. To create an ontology, the snippet information is defined in five sections, which are: – Annotation properties

– Data Properties

– Classes

– Object Properties

– Individuals

Here, object properties are responsible for communicating between entities. In this way, the status of simultaneous change on entities is determined. EdgeLoop, for example, is linked to FaceBound by the HasEdgeLoop property as illustrated in Fig. 6.

Figure 6 The snippet information ontology.

On the other hand, some data can be edited simultaneously, and some cannot. Thus, in Data Properties, two categories are considered as modifiable and unmodifiable as illustrated as an example in Fig. 7.

Figure 7 Modifiable and unmodifiable snippet data.

Mechanism

Therefore, if an entity was associated with another entity through object properties, that secondary entity could not be changed simultaneously. Also, if an entity was associated with a value through data properties that is unmodifiable, it will not be able to change simultaneously.

Finally, one can use the SPARQL (SPARQL Protocol and RDF Query Language) semantic web language, which is similar to relational languages, to find interactions. Use this language to define rules. The inference engine used is also Pellet. This research applied the Snap SPARQL version, which can also use the inferred ontology as shown as sample codes in Table 4. For each entity, a property called Is-Under-Development of Boolean type is considered. This property, like a token, becomes true for any changing entity.

Table 4 The showcase code of the related sections.

PREFIX rdf: <http://www.w3.org/1999/02/22-rdf-syntax-ns#>	
PREFIX owl: <http://www.w3.org/2002/07/owl#>	
PREFIX rdfs: <http://www.w3.org/2000/01/rdf-schema#>	
PREFIX xsd: <http://www.w3.org/2001/XMLSchema#>	
PREFIX :<http://www.semanticweb.org/amin/ontologies/STEP_manufacturing#>	
##################STEP 0 find all targets	
SELECT distinct ?value ?property ?x ?property2 ?value2	
WHERE {	
?property a owl:ObjectProperty.	
?value ?property ?x.	
?property2 a owl:ObjectProperty.	
?x ?property2 ?value2.	
}	

Related entities in the first level are locked with the specified developing entity as shown in the sample codes in Table 5.

Table 5 Check-in code in developing section.

PREFIX rdf: <http://www.w3.org/1999/02/22-rdf-syntax-ns#>	
PREFIX owl: <http://www.w3.org/2002/07/owl#>	
PREFIX rdfs: <http://www.w3.org/2000/01/rdf-schema#>	
PREFIX xsd: <http://www.w3.org/2001/XMLSchema#>	
PREFIX :<http://www.semanticweb.org/amin/ontologies/STEP_manufacturing#>	
######################STEP 1 check if this instance is underdevelopmentt	
SELECT distinct *	
WHERE {	
:EDGE_CURVE_72 :Is_Under_Development ?value.	
filter(?value=true)	
}	

Areas of the developing entity itself that can be changed simultaneously are identified as shown in the sample codes in Table 6.

Table 6 The showcase code of the parts that can be changed at the same time.

PREFIX rdf: <http://www.w3.org/1999/02/22-rdf-syntax-ns#>	
PREFIX owl: <http://www.w3.org/2002/07/owl#>	
PREFIX rdfs: <http://www.w3.org/2000/01/rdf-schema#>	
PREFIX xsd: <http://www.w3.org/2001/XMLSchema#>	
PREFIX :<http://www.semanticweb.org/amin/ontologies/STEP_manufacturing#>	
#####################STEP 2 find unmodifiable related instances	
SELECT distinct ?dataProperty ?dataValue ?domainProperty ?domainValue ?rangeValue	
?rangeProperty	
WHERE {	
?domainProperty a owl:ObjectProperty.	
:EDGE_CURVE_72 ?domainProperty ?domainValue.	
?rangeProperty a owl:ObjectProperty.	
?rangeValue ?rangeProperty :EDGE_CURVE_72.	
?dataProperty rdfs:subPropertyOf :unmodifiable.	
:EDGE_CURVE_72 ?dataProperty ?dataValue	
}	
PREFIX rdf: <http://www.w3.org/1999/02/22-rdf-syntax-ns#>	
PREFIX owl: <http://www.w3.org/2002/07/owl#>	
PREFIX rdfs: <http://www.w3.org/2000/01/rdf-schema#>	
PREFIX xsd: <http://www.w3.org/2001/XMLSchema#>	
PREFIX :<http://www.semanticweb.org/amin/ontologies/STEP_manufacturing#>	
#####################STEP 3 find modifiable related instances	
SELECT *	
WHERE {	
?dataProperty rdfs:subPropertyOf :modifiable.	
:EDGE_CURVE_72 ?dataProperty ?dataValue	
}	

Accordingly, if two designers want to make changes to a section, while notifying the latter user, changes are possible for the former as shown in Fig. 8. According to the queue rule, after the changes are finalized by the former user, the latter one must merge the changes with the first user and then finalize. Once the design is finalized, the snippet will be released.

Figure 8 Behavior of two users on a snippet.

Discussion

Considering the requirements for collaborative product development environment, this article has established the basis for a collaborative product design environment. The main merits of the proposed platform can be discussed as: Enabling the globalized collaboration of product designers for working mutually and increase the efficiency and decreasing the lead times of product design through parallel access and collaboration. This is achieved by the developed ontology framework for managing the product design information contents and proposing the logic analysis for design creation and modification activities.

Fulfilling the requirements of integrated product data by both creating a STEP standard based data model and also developing an ontology model to avoid the conflicts of product design modifications in parallel collaboration. This is achieved by creation of semantic mapping from the RDF semantics with the STEP standard data model. The integrated ontology model consists of the semantic units compatible with STEP standard application protocol for product design data integration.

Enabling the real time interaction of designers through the developed ontology model and its compatibility with XML and RDF based data serialization models. This is achieved with the connection of proposed STEP XML data containers to the developed ontology model. The data communication with design interfaces is facilitated for their collaboration over the globe and also the management and tracking the design modifications.

Supporting the interface seamless connection with computer aided design software packages. This will enable the interoperability of commercial design software packages in real time mode.

It also worth mentioning that the developed ontology framework provides the interoperability of other discipline solutions like artificial intelligence-based framework for accessing the collaboration historical data and design contents for creating more added values. This includes the data analytics applications for creation of knowledge and insights about the possible scenarios of efficiency increase in product design collaborations and efficient strategies for avoiding of conflicts. Based on recent developments in this topic like (Jian et al., 2023), Our work has the potential to facilitate collaboration among engineers in different sectors, enabling the application of the flagging method not only to geometric data but also to other aspects of the product life cycle.

Moreover, the developed solution can be extended for wider disciplines of collaboration for product design and development like product engineering and manufacturing. The possibility of using the IoT paradigm for seamless and real time data communication from manufacturing layer in a collaborative environment will be efficiently possible by extending the proposed ontology model.

Conclusions

Nowadays, globalization, industrial development, and technological advancement have brought about dramatic changes in various sectors of production. Intense competition manufacturers and firms experience when designing new products ranging from product increase and quality improvement to cost reduction intensifying day by day. Therefore, need to achieve new patterns in the design, development and production of products globally. Research centers have done or are doing a lot of research in this area; however, it does not seem to be a very effective solution so far.

By studying in the field of product design and development, in an integrated and collaborative approach, the article seeks to provide solutions at the global manufacturing level. Integrating operations, participatory design and development, and decentralized distribution network, is one of the goals to be achieved by providing a platform.

This study first examines the trend of industrial revolutions and the history of different production systems and then presents a model for product design and development in a participatory and integrated manner. Integrity is also based on the STEP standard. The proposed model is then redefined and upgraded based on the cloud-based cloud generation model. Finally, this model is implemented on a piece and is reviewed as a case study.

The proposed platform provides a solution for both developers and product designers. The purpose of this architecture is to provide a solution to promote collaborative design on parts of the work format of designers in any geographical location. This article in its proposed architecture first allows designers to work and develop their product with the tools they want, then by using XML on the one hand and the STEP standard and its protocols especially with to be used On the AP242 protocol, we achieve the STEP-XML format, thus ensuring integrity at this level. In order to control interferences in the joint design of this article, ontology, which is a method based on artificial intelligence, is used. For this purpose, STEP-XML is converted to RDF and finally OWL with the considered converters. With the creation of OWL and based on ontology, it has entered the architectural logic section, where possible interactions are investigated. Therefore, it is possible to design simultaneously. Finally, in the storage section, all new changes are stored in tables along with previous information based on date and user. The storage of all work steps and the history of activities provides the possibility of retrieving and accessing information at any time and place. It should be noted that the proposed model is redefined and upgraded based on the cloud production model in a service-oriented manner, in such a way that it is possible to share, design at the same time and increase speed, improve quality and reduce costs. Based on this, the presented modeler can be accessed only by connecting to any part of the world with minimal computer equipment.

The case study based on the proposed model demonstrates its capability and indicates that the needs of the product design and development sector are met in a participatory, distributed and integrated manner in the direction of globalization. The proposed model that uses cloud computing has an acceptable security in the data retention process due to the existence of cloud computing data protection protocols. The future studies are encouraged for extending the proposed ontology model for encompassing further product design and development disciplines as well as using the possibility of data analytics module for setting strategies and guidelines for effective collaborative product development using the knowledge regarding conflict scenarios.

Additional Information and Declarations

Competing Interests

Author Contributions

Data Availability

The authors declare that they have no competing interests.

SeyedehTina Zaringhalam conceived and designed the experiments, performed the experiments, performed the computation work, prepared figures and/or tables, authored or reviewed drafts of the article, and approved the final draft.

Mohammad Khalilzadeh analyzed the data, authored or reviewed drafts of the article, and approved the final draft.

Omid Fatahi Valilai conceived and designed the experiments, analyzed the data, prepared figures and/or tables, authored or reviewed drafts of the article, and approved the final draft.

The following information was supplied regarding data availability:

All the developed software modules and codes are available at Figshare:

Fatahi Valilai, Omid (2022). STEP standard interprator. figshare. Software. https://doi.org/10.6084/m9.figshare.21401163.v1.

All the case studies and the related data sets are also available at Figshare:

Fatahi Valilai, Omid (2022). Sample STEP file. figshare. Dataset. https://doi.org/10.6084/m9.figshare.21401262.v1.

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
