# Peer review of "An interactive and integrated framework for collaborative product development in cloud manufacturing using STEP standard-based ontology model"

_PeerJ Computer Science, doi:10.7717/peerj-cs.1530_

## Round 0.1 · original submission · Major Revisions

Please consider the reviewers' comments when resubmitting the paper. Thank you for considering PeerJ Computer Science to publish your research paper.

Reviewer 1 ·

Basic reporting

This paper aims to research STEP standard-based data integrity. The authors propose an interactive
and integrated framework for a cloud collaborative manufacturing system. The subject of the paper is interesting.However,regarding my review, I found that this paper is not innovative enough. What is the research question? Accordingly, what is the innovation and main contributions? In this paper,no specific algorithm is proposed for a certain type of data exchange. In a nutshell, the work is a simple extension of previous work and has very little technical merit. I think this paper is more like a technical report.

Experimental design

no comment

Validity of the findings

no comment

Additional comments

no comment

Reviewer 2 ·

Basic reporting

no comment

Experimental design

Introduction does not clearly talks about how the proposed approach is helping solving the problem space.It also does not talk about the STEP standard which is key of the paper.

Validity of the findings

Conclusions does not clarify on impact of the proposed approaches.

Reviewer 3 ·

Basic reporting

no comment

Experimental design

no comment

Validity of the findings

no comment

Additional comments

The discussion about the cloud manufacturing can be included not only cloud computing.

Reviewer 4 ·

Basic reporting

no comment

Experimental design

The article presents the solution using the STEP exchange file, for which consider it essential to increase the number of references with solutions based on the ISO10303 standard, as well as the information regarding the construction and implementation of the solution; to know if mapping tables, EXPRESS definitions, Application Interpreted Model (AIM) were used, important aspects indicated by the standard.

It is necessary to expand the information regarding integration, specifically in the "mapping layer" section, how integration is projected (CAD/CAPP/CAM/CAI) and the scope achieved for CAx formats. Differences in application protocols AP203, AP214, AP238, AP219 were considered ?.

Figure 2 can be complemented with a description of the features included in the design in relevant terms for CAx systems, such as (STEP-BASED FEATURE) or (Model-Based definition).

Validity of the findings

no comment

Additional comments

The research is novel and of significant impact on developing new solutions in the line of intelligent manufacturing. It presents a clear contribution that can be the basis for future work projected to overcome integration and interoperability barriers using neutral exchange files.

Excellent work!

---

## Round 0.2 · Major Revisions

Please, consider the comments of reviewer 1 to improve your paper so that it can be considered for publication in the journal PeerJ Computer Science journal.

Reviewer 1 ·

Basic reporting

Nothing

Experimental design

Nothing

Validity of the findings

Nothing

Additional comments

As a reversion R1, this version is not satisfactory for solving the original problems.
1. The ontology-based data integrity mehtod with XML-based and OWL-based STEP has already in use for over a decade.
(please refer to: XOEM plus OWL-based STEP product information uniform description and implementation, Journal of Networks, v 6, n 12, p 1662-1667, 2011.)
2. I can't find about STEP-based feature modeler what the authors proposed in the abstact.
(About STEP semantic feature,please refer to : An improved NBA-based STEP design intention feature recognition,Future Generation Computer Systems, 2018,88(6):357-362. DOI: 10.1016/j.future.2018.05.033.)
3. As we know ,STEP AP242 contains PMI information, but I can't find any PMI information in the STEP AP242-based XML or OWL format in this manuscript.
(please refer to: QSCC: A Quaternion Semantic Cell Convolution Graph Neural Network for MBD Product Model Classification,IEEE Transactions on Industrial Informatics,2023,DOI: 10.1109/TII.2023.3246066.

This research has practical application value. It needs a lot of basic work, the workload is very large. I think it is acceptable before the authors can revise it carefully. Good lucky!

Reviewer 3 ·

Basic reporting

The authors met all my previous comments.

Experimental design

The authors met all my previous comments.

Validity of the findings

The authors met all my previous comments.

Additional comments

The authors met all my previous comments.

---

## Round 0.4 · accepted · Accept

Thank you for considering the reviewers' comments and making the required changes to your article.
I am happy to inform you that your paper now reaches the required level for publication in PeerJ Computer Science. Thank you for considering our journal for publishing your research papers.
We hope you will continue to consider the journal for publication in your future research.

Sincerely,

M. Emilia Cambronero
Academic Editor of Peer J Computer Science

Reviewer 1 ·

Basic reporting

Although I can't find the responsetoreviewer file, I think this version is acceptable because of the authors' lots of work.

Experimental design

no comment

Validity of the findings

no comment